# Excellent and Good Results Treating Stiffness with Early and Late Manipulation after Unrestricted Caliper-Verified Kinematically Aligned TKA

**DOI:** 10.3390/jpm12020304

**Published:** 2022-02-18

**Authors:** Adithya Shekhar, Stephen M. Howell, Alexander J. Nedopil, Maury L. Hull

**Affiliations:** 1California Northstate University School of Medicine, Elk Grove, CA 95758, USA; adithya.shekhar6487@cnsu.edu; 2Department of Biomedical Engineering, University of California, Davis, CA 95616, USA; mlhull@ucdavis.edu; 3Adventist Health Lodi Memorial, Lodi, CA 95240, USA; nedopil@me.com; 4Orthopädische Klinik König-Ludwig-Haus, Lehrstuhl für Orthopädie der Universität, 97074 Würzburg, Germany; 5Department of Mechanical Engineering, University of California Davis Medical Center, Sacramento, CA 95817, USA

**Keywords:** reoperation, revision, implant survival, forgotten joint score, Oxford knee score

## Abstract

Manipulation under anesthesia (MUA) for stiffness within 6 to 12 weeks after mechanically aligned total knee arthroplasty (TKA) generally yields better outcome scores than an MUA performed later. However, the timing of MUA after unrestricted, caliper-verified, kinematically aligned (KA) TKA remains uncertain. A retrospective review identified 82 of 3558 (2.3%) KA TKA patients treated with an MUA between 2010 and 2017. Thirty patients treated with an MUA within 3 months of the TKA (i.e., early) and 24 in the late group (i.e., >3 months) returned a questionnaire after a mean of 6 years and 5 years, respectively. Mean outcome scores for the early vs. late group were 78 vs. 62 for the Forgotten Joint Score (FJS) (*p* = 0.023) and 42 vs. 39 for the Oxford Knee Score (OKS) (*p* = 0.037). Subjectively, the early vs. late group responses indicated that 83% vs. 67% walked without a limp, 73% vs. 54% had normal extension, and 43% vs. 25% had normal flexion. An MUA within 3 months after unrestricted KA TKA provided excellent FJS and OKS at final follow-up relative to a late MUA. A late MUA performed after 3 months is worth consideration because of the good FJS and OKS scores, albeit with a risk of a persistent limp and limitation in knee extension and flexion.

## 1. Introduction

Stiffness after total knee arthroplasty (TKA) is a multifactorial complication, occasionally necessitating manipulation under anesthesia (MUA) or revision surgery [1]. The complaints reported by patients with stiffness are an inadequate range of motion (ROM) and limitations in functional activities, despite a trial of postoperative self-administered exercise or a formal physical therapy program.

The reported incidence of MUA for stiffness after mechanically aligned (MA) TKA ranges from 1 to 12% [2,3,4]. Factors contributing to the risk of stiffness include component malposition, patients being younger, smoking, prior knee surgery, diabetes, use of anticoagulation drugs (i.e., warfarin), limited preoperative ROM, and a long tourniquet time [4,5,6,7,8].

The timing of performing an MUA is an essential consideration. Patients treated within 6 to 12 weeks after the primary TKA report better improvements in final ROM than those treated later [4,7]. Patients who undergo an MUA after primary MA TKA are at higher risk of revision surgery and worse long-term clinical outcome scores, ROM, and implant survivorship relative to those that did not undergo MUA [2].

There are quantitative guidelines for determining whether differences in patient-reported outcome scores between treatments, such as the timing of an MUA, are clinically important and excellent or good. For example, an eight-point difference for the Forgotten Joint Score (100 best, 0 worst) between treatments is clinically important, and values of 80 and 70 points indicate none and minor restrictions in knee function, respectively [9,10]. For the Oxford Knee Score (48 best, 0 worst), a five-point difference between treatments is clinically important, and scores in the range of 42–48 and 34–41 points are considered excellent and good results, respectively [11,12,13]. In addition, the patient can subjectively report whether the MUA resolved their limp and restored normal, nearly normal, or abnormal knee extension and flexion.

Unrestricted, caliper-verified kinematic alignment resurfaces the knee by making the thickness of the distal and posterior femoral resections match those of the condyles of the femoral component, after adjusting 2 mm for missing cartilage and 1 mm for the kerf of the blade. The tibial resection matches the varus–valgus angle of the pre-arthritic knee when a spacer block and trial components create a tight rectangular extension space. The ultimate goal is to co-align the transverse flexion–extension axes of the tibiofemoral and patellofemoral prosthetic joints parallel to the pre-arthritic joint lines [14]. Because KA ignores the femoral and ankle centers, MA technology, such as robotic and navigational instrumentation, is unnecessary. These technologies perform the femoral resections less accurately than the manual instruments that directly reference the distal and posterior femur [15].

Although an early (i.e., within 3 months) MUA after MA TKA generally gives better results, no studies have evaluated the effect of the timing of the MUA on patient satisfaction and long-term outcomes after an unrestricted caliper-verified KA TKA. Therefore, the present study determined whether an MUA performed early, within 3 months, results in better clinical outcome scores, higher patient-reported satisfaction and fewer revision surgeries than one performed later.

## 2. Materials and Methods

An Institutional Review Board (Ref. No. Pro00049603) approved this retrospective cohort analysis from our prospectively collected electronic database of 3558 primary unrestricted caliper-verified KA TKAs performed by a single surgeon between 2010 and 2017. During this treatment interval, a total of 82 patients underwent an MUA for knee stiffness. At the time of primary TKA, each patient fulfilled the medical necessity guideline of the Centers for Medicare and Medicaid Services for TKA treatment. Included were osteoarthritic knees with (1) radiographic evidence of Kellgren–Lawrence Grade III to IV arthritic change or osteonecrosis; (2) any severity of clinical varus or valgus deformity; and (3) any severity of flexion contracture. A single surgeon (SMH) performed unrestricted caliper-verified KA TKA using manual instruments through a 1/3rd medial vastus approach. Caliper measurements enabled the matching of the thickness of the femoral bone resections to the thickness of the femoral component within ±0.5 mm after correcting for wear and the saw blade’s kerf using a previously described technique [14]. Postoperatively, patients underwent physical therapy.

Patients with either a 10° flexion contracture or flexion less than 90° that failed to improve after a 3-week trial of prone extension and sitting flexion stretching exercises were offered an MUA. The MUA was performed on an outpatient basis in the hospital or in an ambulatory surgery center by the same surgeon that performed the primary TKA. Under anesthesia, the knee was injected with 30 cc of 0.5% bupivacaine and 40 mg of Depo-Medrol. The knee was gently brought into terminal extension and then into deep flexion to stretch adhesions.

Each patient was grouped based on the time between the MUA and the TKA. Patients treated within 3 months were assigned to the early group and the remaining were assigned to the late group. The CONSORT diagram shows the number of patients assessed for eligibility, the number of excluded patients and the reason, and the number of patients in the early and late groups that participated in the study (Figure 1). One author (AS) extracted pre-TKA and pre-MUA patient characteristics, clinical outcome scores, and implant design from the prospectively collected digital database. Between July 2020 and December 2020, one observer (AS), independently from the treating surgeon, contacted each patient by e-mail, postal service, or phone. For those patients with outdated contact information, current whereabouts were gathered by querying five “people search” websites. The patients were sent a questionnaire asking them to complete the Forgotten Knee Score (100 best, 0 worst) and Oxford Knee Score (OKS) (48 best, 0 worst). In addition, the questionnaire asked whether they had a reoperation on the TKA after the MUA and whether their final gait, extension, and flexion were normal, nearly normal, or abnormal. For those that underwent a post-MUA reoperation, a review of the operative note determined the type of corrective surgery.

Discrete variables (patient-reported outcomes) were reported as number (percentage) (JMP Pro, 15.0.0, http://www.jmp.com, access date 12 January 2022). Continuous variables were reported as either the mean ± standard deviation (SD) or the median and interquartile range (IQR) depending on the normality of the data. A Wilcoxon/Kruskal–Wallis test determined the significance of the difference in the FJS and OKS between early and late MUA groups. Significance was *p* < 0.05.

## 3. Results

Before the primary TKA, there were no differences in the sex, age, body mass index, smoking status, diabetic status, and clinical outcome scores between the 30 patients in the early group (MUA within 3 months) and the 24 patients in the late group (MUA after 3 months) (Table 1). Of the 54 patients, 51 had a cemented posterior cruciate ligament-retaining implant, two had a posterior cruciate ligament-substituting implant, and one had a posterior stabilized implant. No patients were revised for stiffness without first undergoing an MUA. Few patients offered the MUA declined to undergo the procedure. The primary reason for MUA was loss of flexion.

The mean time interval between the primary KA TKA and the MUA was 2 ± 0.6 months, which was shorter than the 7 ± 5.6 months for the late group (*p* < 0.0001). At the time of the MUA, the early group had less knee extension and flexion, a lower Knee Society Score and Knee Function Score (*p* = 0.047, *p* = 0.000, *p* = 0.013, and *p* = 0.005, respectively), and a higher mean OKS (*p* = 0.016) relative to the late group (Table 2).

The mean time interval between the MUA and the final follow-up for the early group was 6 ± 2.2 years, which was comparable to the 5 ± 1.6 years for the late group (NS). The early group had a 14-point higher mean FJS and a 3-point higher OKS than the late group (*p* = 0.049 and *p* = 0.025, respectively) (Table 3). A higher percentage of patients in the early group walked without a limp and had normal knee extension and flexion relative to the late group. One patient in the early and one in the late group underwent arthroscopic lysis of adhesions and lateral release. One patient in the late group underwent a two-stage revision for stiffness and tibial loosening for suspected infection. Subsequent pathology and culture reports indicated that the cause of the loosening was an unsuspected lymphoma in the proximal tibia and not an infection.

## 4. Discussion

The present case series examined patients that underwent an MUA after unrestricted caliper-verified KA TKA to determine whether an early MUA (i.e., within 3 months) resulted in better clinical outcome scores, higher patient-reported satisfaction, and fewer revision surgeries than an MUA performed later. Two significant findings could aid decision making and help manage patients’ expectations. The first was that an early MUA provided excellent FJS and OKS relative to a late MUA. The second was that a late MUA resulted in good FJS and OKS, albeit with a risk of a persistent limp and limitations in knee extension and flexion.

The timing of the MUA is hotly debated, with MA studies recommending performing it within 3 months after TKA [13]. A comparison of the present study’s early vs. the late group’s mean FJS (78 vs. 62) and mean OKS (42 vs. 39) at final follow-up of 6 vs. 5 years after MUA to current reports of these outcome scores after restricted KA TKA and MA TKA without MUA can help clarify the effectiveness of the timing of the MUA. A study comparing restricted primary KA TKA performed with robotic arm technology and MA TKA performed with manual instruments reported a mean FJS of 72 and 61 in patients without an MUA at 18-month follow-up, respectively [16]. In addition, a study of the OKS after MA TKA without an MUA reported a mean OKS of 32 at 12-month follow-up [17]. Hence, the FJS and OKS of the early and late MUA groups in the present study are comparable to or better than recent studies of restricted KA TKA and MA TKA. Therefore, the surgeon and patient have leeway when considering the timing of the MUA after unrestricted KA TKA, since those treated at a mean of 7 months had good outcome scores relative to MA TKA.

The surgeon could have offered those patients with a poor pre-MUA function in the late group an earlier MUA. Therefore, it is interesting to know whether those in the late group with poor pre-MUA function fared worse than those with better pre-MUA function. Accordingly, a post-op analysis of the late MUA group assigned the 11 patients with a pre-MUA Oxford Knee <30 points to the poor function subgroup and the 12 with a pre-MUA Oxford Knee >30 points to the better function subgroup. The mean improvement between the pre-MUA and final Oxford Knee Score for the poor function subgroup was 19 points (21 to 40) and was greater than the improvement of 6 points (34 to 44) for the better function subgroup (*p* = 0.0226). Hence, not offering an earlier MUA to those patients with poor function as measured by the Oxford Knee Score did not diminish the effectiveness of the MUA.

The surgical technique and patient characteristics in the present study might explain the excellent and good clinical outcome scores after the early and late MUAs. First, component malposition was unlikely, as unrestricted caliper-verified KA sets the components within 0 ± 0.5 mm of the patient’s pre-arthritic femoral and tibial joint lines. Components set coincident to the native joint surface restore the resting lengths of the collateral and posterior cruciate ligaments, which restores native medial and lateral tibial compartment forces without the morbidity of ligament release [18,19,20,21,22,23]. Resurfacing the knee reduces the risk of kinematic conflict from component malposition, enabling the MUA to regain motion. Second, although the mean age of 61 and 65 years for the early and late groups was younger relative to most cohorts of patients treated with TKA and 20 of 54 had had prior knee surgery, few patients smoked (2 of 54) or had diabetes (4 of 54). Finally, aspirin at 81 mg BID reduced the risk of thromboembolic events in 47 patients. Only the seven patients preoperatively prescribed anticoagulants more potent than aspirin took them postoperatively.

The 2.3% (82 of 3558) of patients that underwent an MUA after unrestricted caliper-verified KA TKA is comparable to or lower than the incidence of 6.5% (182 of 2783), 4.6% (164 of 3556), 3.6% (62 of 1729) reported after MA TKA performed in a single institution [4,6,9]. Because 2 to 6 out of a 100 TKAs undergo an MUA, the long-term risk of further surgery is of interest to the patient and surgeon [1,2] A registry study of 664,604 primary MA TKAs in which 3918 (0.6%) underwent MUA after a median of 2.0 +/− 1.0 months reported that revision surgery occurred in 131 (3.4%) MUA patients after a median of 9.0 months [1]. A case series of 2193 patients (2783 knees) that underwent primary MA TKA reported that revisions occurred in 18 of 182 knees (9.9%) after the MUA, with the most common reason being continued stiffness [2]. Relative to these MA studies, the present study of unrestricted caliper-verified KA TKA had only one patient in the late MUA group who underwent the removal of implants for tibial loosening, which was caused by an unsuspected lymphoma in the proximal tibia initially diagnosed as an infection. In addition, one patient in the early and one in the late group underwent arthroscopic lysis of adhesions and lateral release. Hence, the present study’s revision/reoperation rate of 6% at 5 to 6 years after an MUA performed early or late after unrestricted caliper-verified KA TKA is comparable to reports of MA TKA.

There were several limitations to this study. First, the decision to proceed with MUA was not guided by a uniform algorithm or standard definition of “unacceptable” postoperative stiffness, but rather by patient input and surgeon judgment. Another is that the patient’s knee extension and flexion was not measured at the final follow-up. As a surrogate for ROM measurements, the patient reported whether knee extension and flexion were normal, nearly normal, or abnormal. This subjective categorization showed that the early MUA group reported better final ROM than the late MUA group. A third is that our cohort was not large enough to identify baseline patient variables associated with the development of a stiff TKA. A fourth limitation is that the effectiveness of the MUA after unrestricted KA did not evaluate all levels of osteoarthritic knee complexity. However, the MUA improved those stiff primary KA TKAs that underwent a prior arthrotomy (24%), arthroscopy (41%), and ACL reconstruction (4%). Finally, one author is also the developer of the KA TKA technique and, therefore, might be prone to bias. The developer of a technique often achieves significantly better results than independent registry studies using the same technique, and this limits the generalization of the findings [24].

## 5. Conclusions

The present study showed that an early (i.e., within 3 months) and late MUA (i.e., longer than 3 months) for stiffness after unrestricted caliper-verified KA TKA provided excellent and good FJS and OKS at 6- and 5-year follow-ups, respectively, with a low risk of reoperation/revision. This information can aid decision making and help manage patients’ expectations.

## Figures and Tables

**Figure 1 jpm-12-00304-f001:**
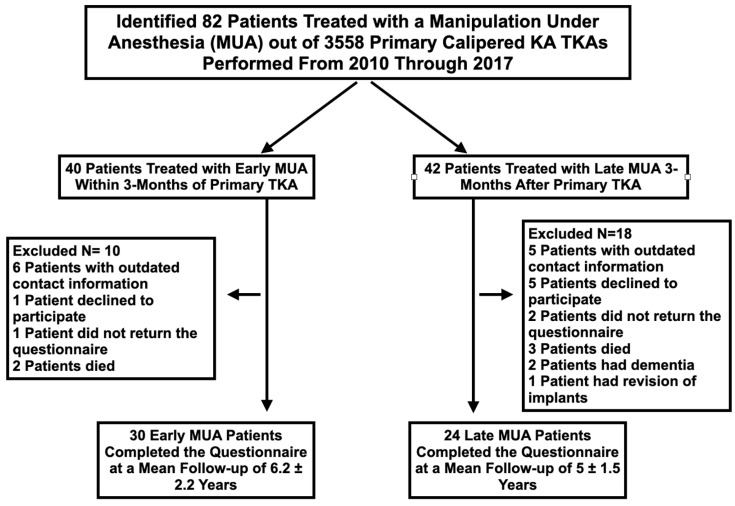
The CONSORT diagram shows, for the early and late groups, the number of patients assessed for eligibility, the number excluded with the specific reason, and the number that completed the questionnaire.

**Table 1 jpm-12-00304-t001:** Patient Characteristics Prior to Primary Calipered KA TKA for Patients Treated with an Early and Late Manipulation Under Anesthesis (MUA).

Patient Characteristics Prior to Primary Caliper Verified KA TKA	Early MUA within 3 Months of Caliper Verified KA TKA	Late MUA > 3 Months after Caliper Verified KA TKA	*p*-Value
Number of Patients that Completed Questionnaire	30	24	–
Male	14 (43%)	10 (46%)	–
Female	16 (57%)	14 (54%)	–
Mean Age (years)	61 ± 8.5	65 ± 5.5	NS
Mean Body Mass Index (kg/m^2^)	28 ± 3.8	29 ± 5.9	NS
Mean Pre-TKA Extension (degrees)	13 ± 7.2	11 ± 7.4	NS
Mean Pre-TKA Flexion (degrees)	112 ± 11.6	113 ± 15.3	NS
Prior knee surgery	14 yes, 16 no	6 yes, 18 no	NS
Smoking	1 (3.2%)	1 (3.3%)	NS
Diabetes	2 (6.5%)	2 (6.7%)	NS
Mean Pre-TKA Oxford Knee Score (48 is best, 0 is worst)	21 ± 7.6	22 ± 6.5	NS
Mean Pre-TKA Knee Society score (100 is best, 0 is worst)	33 ± 13.8	39 ± 18.0	NS
Mean Pre-TKA Knee Function Score (100 is best, 0 is worst)	49 ± 20.0	53 ± 12.0	NS

**Table 2 jpm-12-00304-t002:** Patient Characteristics Pre Manipulation for Patients Treated with an Early and Late MUA.

Patient Characteristics Pre Manipulation under Anesthesia (MUA)	Early MUA within 3 Months of Caliper Verified KA TKA	Late MUA > 3 Months after Caliper Verified KA TKA	*p*-Value
Number of Patients that Completed Questionnaire	30	24	–
Mean Number of Months Between MUA and Primary Caliper Verifed KA TKA	2 ± 0.6	7 ± 5.6	*p* < 0.000
Mean Extension at Time of MUA (degrees)	9 ± 8.4	5 ± 8.3	*p* = 0.047
Mean Flexion at Time of MUA (degrees)	79 ± 15.3	95 ± 12.8	*p* = 0.000
Mean Oxford Knee Score at Time of MUA (48 is best, 0 is worst)	22 ± 7.2	28 ± 8.9	*p* = 0.016
Mean Knee Society score at Time of MUA (100 is best, 0 is worst)	64 ± 23.4	80 ± 21.2	*p* = 0.013
Mean Knee Function Score at Time of MUA (100 is best, 0 is worst)	46 ± 18.7	63 ± 17.2	*p* = 0.005
Reason for MUA			
Loss of Flexion	23 (85%)	20 (74%)	
Loss of Extension	3 (11%)	1 (4%)	
Loss of Extension and Flexion	1 (4%)	6 (22%)	

**Table 3 jpm-12-00304-t003:** Patient Characteristics at Final Follow-up After MUA for Patients Treated with an Early and Late MUA.

Patient Characteristics at Final Follow-Up after Manipulation under Anesthesia (MUA)	Early MUA within 3 Months of Caliper Verified KA TKA	Late MUA > 3 Months after Caliper Verified KA TKA	*p*-Value
Number of Patients that Completed Questionnaire	30	24	
Mean Number of Years Between MUA and Final Follow-up	6 ± 2.3	5 ± 1.5	NS
Mean Post-MUA Oxford Knee Score (48 is best, 0 is worst)	42 ± 8.9	39 ± 8.1	*p* = 0.037
Mean Post-MUA Forgotten Joint Score (100 is best, 0 is worst)	78 ± 28.2	62 ± 31.0	*p* = 0.023
Patients that Walk Without a Limp	83% Yes, 17% No	67% Yes, 33% No	NS
Patients Within a Subjective Category of Knee Extension	73% Normal, 20% Nearly Normal, 7% Abnormal	54% Normal, 42% Nearly Normal, 4% Abnormal	NS
Patients Within a Subjective Category of Knee Flexion	43% Normal, 40% Nearly Normal, 17% Abnormal	25% Normal, 58% Nearly Normal, 17% Abnormal	NS
Patients Treated with a Reoperation after a Failed MUA	1 Arthroscopic Lateral Release with Lysis of Adhesions	1 Arthroscopic Lateral Release with Lysis of Adhesions 1 Revision for tibial loosening caused by lymphoma	N/A

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
