# Peer review of "Excellent and Good Results Treating Stiffness with Early and Late Manipulation after Unrestricted Caliper-Verified Kinematically Aligned TKA"

_jpm, 2022, doi:10.3390/jpm12020304_

Round 1
Reviewer 1 Report
General comments: this paper is relevant for this field, well structured, clearly described and not a repetition of known science; I find that using Forgotten Joint Score (mostly) and Oxford Knee Score in this paper is an excellent choice for studying the proposed hypothesis;
Introduction: provides the necessary background and hypothesis is clearly given;
Material and methods: description of the methods is accurate and the study can be repeated using the description in this section; the procedure described (manipulation under anesthesia)
Results are matching the previous part (Material and methods); they are adequately reported, all being quality tested;
Figures, tables and schemes are possible to read as a standalone part of this manuscript, being clear and simple to understand;
Discussion starts in a proper manner, with the principal findings of the study, all this findings being placed in an appropriate context regarding conclusions of previous studies; limitations of this study are clearly and fairly mentioned;
Conclusions are short and based on data within this paper;
Ethics statement are adequate, clearly defined and most of cited references are within the last 5 years
Study design is appropriate regarding studied hypothesis;
Number of self citations is 6, and I suggest a small reevaluation of number of "the"-s, those being only reasons I suggest a minor review
Author Response
Reviewer 1
Number of self citations is 6, and I suggest a small reevaluation of number of "the"-s, those being onlyreasons I suggest a minor review
We deleted the following two out of six references that were self citations as suggested.
- Shelton TJ, Howell SM, Hull ML (2019) Is There a Force Target That Predicts Early Patient-reported Outcomes After Kinematically Aligned TKA? Clin Orthop Relat Res 477:1200-1207 (54)
- Riley J, Roth JD, Howell SM, Hull ML (2018) Internal-external malalignment of the femoral component in kinematically aligned total knee arthroplasty increases tibial force imbalance but does not change laxities of the tibiofemoral joint. Knee Surg Sports Traumatol Arthrosc 26:1618-1628 (56)
Reviewer 2 Report
Overall this is a well-designed and written paper. Since caliper based technique of kinematically aligned TKAs is not very well described in the literature (yet controversial) I consider it suitable for publication, however the authors need to resolve several major and a few minor problems
Major problems :
- Please provide data regarding the incidence of “stiff knees” / patients who were offered MUA procedures (perhaps some patients declined MUAs). Were there any cases directly revised for stiffness (without MUA) ? Please discuss this with comparison to data for mechanically aligned TKAs
- Please provide more detailed data regarding the indications for MUAs – how many patients had a) limited extension; b)limited flexion; c) limted flexion and extension ? Did this affect the outcome ? I could imagine patients with >10 deg flexion contracture function far better than individuals with 40 deg. Flexion (please discuss / include in limitations)
- You stated that the “late” group had overall better knee function before MUA – please discuss the potential effect of this difference on the outcome – perhaps cases with poorer knee function mobilized later would have poorer ultimate results ? Please discuss / this is also a limitation
- Lines 161-174 – this is a highly controversial and surgeons advocating for mechanical alignment would strongly disagree. This is a good example of bias related to one of authors being the inventor of the technique which you mentioned in the last paragraph – please tone down as there is no consensus on the superiority of kinematic or mechanical alignment.
Some minor issues
Since the authors explained the clinical impact scoring systems (introduction), I would suggest to include a few short remarks briefly explaining
- The difference between mechanical and kinematic alignment
- The difference between the “conventional / navigation based” kinematic alignment and the caliper technique
Please explain who performed the MUA proceudres – the operating surgeon ? One doctor or several doctors ? (In the latter case please discuss this as a potential limitation_
Author Response
Responses to Reviewer Comments
The authors thank the reviewer for taking his/her time to read our manuscript and offer comments to clarify and/or strengthen the presentation.
Responses to Comments of Reviewer 2
Comment 1: Please provide data regarding the incidence of “stiff knees” / patients who were offered MUA procedures (perhaps some patients declined MUAs). Were there any cases directly revised for stiffness (without MUA)? Please discuss this with comparison to data for mechanically aligned TKAs
AUTHORS RESPONSE: We added the following clarifying sentences in the Results and Discussion sections.
RESULTS SECTION: No patients were revised for stiffness without first undergoing an MUA. Few patients who were offered MUA declined to undergo the procedure.
DISCUSSION SECTION: The 2.3% (82 of 3,558) incidence of patients who underwent an MUA after unrestricted caliper-verified KA TKA is comparable to or lower than the incidence of 6.5% (182 of 2,783), 4.6% (164 of 3556), 3.6% (62 of 1729) reported after MA TKA performed in a single institution (Newman, 2018 #5;Knapp, 2020 #14;Crawford, 2021 #43).
Comment 2. Please provide more detailed data regarding the indications for MUAs – how many patients had: a) limited extension; b) limited flexion; c) limited flexion and extension? Did this affect the outcome? I could imagine patients with >10 deg flexion contracture function far better than individuals with 40 deg flexion (please discuss /include in limitations)
AUTHORS RESPONSE: We clarified the cause of the MUA in Table 2 and in the following sentence in the Results section.
RESULTS SECTION: The primary reason for MUA was loss of flexion.
Comment 3. You stated that the “late” group had overall better knee function before MUA – please discuss the potential effect of this difference on the outcome– perhaps cases with poorer knee function mobilized later would have poorer ultimate results? Please discuss / this is also a limitation
AUTHORS RESPONSE: We added the clarification in a new paragraph in the Discussion as follows:
The surgeon could have offered those patients in the late group with a poor pre-MUA function an earlier MUA. Therefore, it is interesting to know whether those in the late group with poor pre-MUA function fared worse than those with better pre-MUA function. Accordingly, a post-op analysis of the late MUA group assigned the 11 patients with a pre-MUA Oxford Knee < 30 points to the poor function subgroup and the 12 with a pre-MUA Oxford Knee > 30 points to the better function subgroup. The mean improvement between the pre-MUA and final Oxford Knee Score for the poor function subgroup was 19 points (21 to 40) and was greater than the improvement of 6 points (34 to 44) for the better function subgroup (p =0.0226). Hence, not offering an earlier MUA to those patients with poor function as measured by the Oxford Knee Score did not diminish the effectiveness of the MUA.
Comment 4. Lines 161-174 – this is highly controversial and surgeons advocating for mechanical alignment would strongly disagree. This is a good example of bias related to one of authors being the inventor of the technique which you mentioned in the last paragraph – please tone down as there is no consensus on the superiority of kinematic or mechanical alignment.
AUTHORS RESPONSE: There is no bias; the statements made are based on the preponderance of the evidence comparing outcomes from KA TKA and MA TKA. To appreciate this evidence, we offer the following list of 13 published randomized trials, case-control studies, and case-series of bilateral TKA with a KA TKA and an MA TKA from authors around the world that compared KA TKA to MA TKA. Eleven showed that one or more outcome variables were better for KA TKA than MA TKA. To our knowledge no published studies have shown that MA TKA had better outcome values than KA TKA.
- Elbuluk, A.M.; Jerabek, S.; Suhardi, J.; Sculco, P.; Ast, M.; Vigdorchik, J.M. Head-to-Head Comparison of Kinematic Alignment versus Mechanical Alignment for Total Knee Arthroplasty. J Arthroplasty 2022, doi:10.1016/j.arth.2022.01.052.
- Yaron, B.Z.; Ilan, S.; Tomer, K.; Eran, B.; Gabriel, A.; Noam, S. Patients undergoing staged bilateral knee arthroplasty are less aware of their kinematic aligned knee compared to their mechanical knee. J Orthop 2021, 23, 155-159, doi:10.1016/j.jor.2020.12.032.
- Niki, Y.; Nagura, T.; Kobayashi, S.; Udagawa, K.; Harato, K. Who Will Benefit From Kinematically Aligned Total Knee Arthroplasty? Perspectives on Patient-Reported Outcome Measures. J Arthroplasty 2020, 35, 438-442 e432, doi:10.1016/j.arth.2019.09.035.
- McEwen, P.J.; Dlaska, C.E.; Jovanovic, I.A.; Doma, K.; Brandon, B.J. Computer-Assisted Kinematic and Mechanical Axis Total Knee Arthroplasty: A Prospective Randomized Controlled Trial of Bilateral Simultaneous Surgery. J Arthroplasty 2020, 35, 443-450, doi:10.1016/j.arth.2019.08.064.
- MacDessi, S.J.; Griffiths-Jones, W.; Chen, D.B.; Griffiths-Jones, S.; Wood, J.A.; Diwan, A.D.; Harris, I.A. Restoring the constitutional alignment with a restrictive kinematic protocol improves quantitative soft-tissue balance in total knee arthroplasty: a randomized controlled trial. Bone Joint J 2020, 102-B, 117-124, doi:10.1302/0301-620X.102B1.BJJ-2019-0674.R2.
- Jeremic, D.V.; Massouh, W.M.; Sivaloganathan, S.; Rosali, A.R.; Haaker, R.G.; Riviere, C. Short-term follow-up of kinematically vs. mechanically aligned total knee arthroplasty with medial pivot components: A case-control study. Orthop Traumatol Surg Res 2020, 106, 921-927, doi:10.1016/j.otsr.2020.04.005.
- Shelton, T.J.; Gill, M.; Athwal, G.; Howell, S.M.; Hull, M.L. Outcomes in Patients with a Calipered Kinematically Aligned TKA That Already Had a Contralateral Mechanically Aligned TKA. The journal of knee surgery 2021, 34, 87-93, doi:10.1055/s-0039-1693000.
- An, V.V.G.; Twiggs, J.; Leie, M.; Fritsch, B.A. Kinematic alignment is bone and soft tissue preserving compared to mechanical alignment in total knee arthroplasty. The Knee 2019, 26, 466-476, doi:10.1016/j.knee.2019.01.002.
- Young, S.W.; Walker, M.L.; Bayan, A.; Briant-Evans, T.; Pavlou, P.; Farrington, B. The Chitranjan S. Ranawat Award : No Difference in 2-year Functional Outcomes Using Kinematic versus Mechanical Alignment in TKA: A Randomized Controlled Clinical Trial. Clin Orthop Relat Res 2017, 475, 9-20, doi:10.1007/s11999-016-4844-x.
- Matsumoto, T.; Takayama, K.; Ishida, K.; Hayashi, S.; Hashimoto, S.; Kuroda, R. Radiological and clinical comparison of kinematically versus mechanically aligned total knee arthroplasty. Bone Joint J 2017, 99-B, 640-646, doi:10.1302/0301-620X.99B5.BJJ-2016-0688.R2.
- Calliess, T.; Bauer, K.; Stukenborg-Colsman, C.; Windhagen, H.; Budde, S.; Ettinger, M. PSI kinematic versus non-PSI mechanical alignment in total knee arthroplasty: a prospective, randomized study. Knee Surg Sports Traumatol Arthrosc 2017, 25, 1743-1748, doi:10.1007/s00167-016-4136-8.
- Waterson, H.B.; Clement, N.D.; Eyres, K.S.; Mandalia, V.I.; Toms, A.D. The early outcome of kinematic versus mechanical alignment in total knee arthroplasty: a prospective randomised control trial. Bone Joint J 2016, 98-B, 1360-1368, doi:10.1302/0301-620X.98B10.36862.
- Dossett, H.G.; Estrada, N.A.; Swartz, G.J.; LeFevre, G.W.; Kwasman, B.G. A randomised controlled trial of kinematically and mechanically aligned total knee replacements: two-year clinical results. Bone Joint J 2014, 96-B, 907-913, doi:10.1302/0301-620X.96B7.32812.
In addition, the following registry study and randomized trial showing that the 7-year survivorship, and the 2-year risk of tibial component migration of KA TKA and MA TKA are comparable;
- Klasan, A.; de Steiger, R.; Holland, S.; Hatton, A.; Vertullo, C.J.; Young, S.W. Similar Risk of Revision After Kinematically Aligned, Patient-Specific Instrumented Total Knee Arthroplasty, and All Other Total Knee Arthroplasty: Combined Results From the Australian and New Zealand Joint Replacement Registries. The Journal of arthroplasty 2020, doi:10.1016/j.arth.2020.05.065.
- Laende, E.K.; Richardson, C.G.; Dunbar, M.J. A randomized controlled trial of tibial component migration with kinematic alignment using patient-specific instrumentation versus mechanical alignment using computer-assisted surgery in total knee arthroplasty. Bone Joint J 2019, 101-B, 929-940, doi:10.1302/0301-620X.101B8.BJJ-2018-0755.R3.
Hence, the literature supports retaining appropriately referenced sentences contained in lines 161-174.
Some minor issues
Since the authors explained the clinical impact coring systems (introduction), I would suggest to include a few short remarks briefly explaining:
The difference between mechanical and kinematic alignment
The difference between the “conventional /navigation based” kinematic alignment and the caliper technique
AUTHORS RESPONSE: We added the clarification in a new paragraph in the Introduction as follows:
Unrestricted caliper-verified kinematic alignment resurfaces the knee by making the thickness of the distal and posterior femoral resections match those of the condyles of the femoral component after adjusting 2 mm for missing cartilage and 1 mm for the kerf of the blade. The tibial resection matches the varus-valgus angle of the pre-arthritic knee when a spacer block and trial components create a tight rectangular extension space. The ultimate goal is to co-align the transverse flexion-extension axes of the tibiofemoral and patellofemoral prosthetic joints parallel to the pre-arthritic joint lines [15]. Because KA ignores the femoral and ankle centers, MA technology such as robotic and navigational instrumentation is unnecessary. These technologies perform the femoral resections less accurately than manual instruments that directly reference the distal and posterior femur {Howell, In Press #60}.
Please explain who performed the MUA procedures– the operating surgeon ? One doctor or several doctors? (In the latter case please discuss this as a potential limitation)
AUTHORS RESPONSE: We added the clarification that the same surgeon that performed the MUA performed the primary TKA as follows:
The MUA was performed on an outpatient basis in the hospital or in an ambulatory surgery center by the same surgeon that performed the primary TKA.
Round 2
Reviewer 2 Report
I appreciate the effort the authors took to correct the manuscript. I really like their response to the part regarding discussion where I suggested toning down the part regarding the superiority of KA vs MA. Still I would like to raise the point that in certain cases – predominantly difficult knees (ex. post-traumatic, rheumatic knees with necrotic lesions, MED etc. ). Consequently I would suggest the authors to include a statement that KA has documented advantages but also its limitations.
Author Response
Responses to Reviewer 2's Comments
The authors thank the reviewer for taking his/her time to read our manuscript and offer comments to clarify and/or strengthen the presentation.
Responses to Comments of Reviewer 2
Comment 1: I appreciate the effort the authors took to correct the manuscript. I really like their response to the part regarding discussion where I suggested toning down the part regarding the superiority of KA vs MA. Still I would like to raise the point that in certain cases – predominantly difficult knees (ex. post-traumatic, rheumatic knees with necrotic lesions, MED etc. ). Consequently I would suggest the authors to include a statement that KA has documented advantages but also its limitations.
AUTHORS RESPONSE: We added the following sentence in the Limitation subsection of the Discussion:
A fourth limitation is that the effectiveness of the MUA after unrestricted KA did not evaluate all levels of osteoarthritic knee complexity. However, the MUA improved those stiff primary KA TKAs that underwent a prior arthrotomy (24%), arthroscopy (41%), and ACL reconstruction (4%).